# Scoping review of qualitative studies on family planning in Uganda

Julie M. Buser[1]*, Pebalo F. Pebolo[2], Ella August[3,4‡], Gurpreet K. Rana[5‡], Rachel Gray[1‡], Faelan E. Jacobson-Davies[1,6‡], Edward Kumakech[7‡], Tamrat Endale[1‡], Anna Grace Auma[7‡], Yolanda R. Smith[1,6‡]

1 Center for International Reproductive Health Training (CIRHT), University of Michigan, Ann Arbor, Michigan, United States of America, 2 Department Reproductive Health, Gulu University Faculty of Medicine, Gulu, Uganda, 3 Department of Epidemiology, University of Michigan School of Public Health, Ann Arbor, Michigan, United States of America, 4 PREPSS (Pre-Publication Support Service), University of Michigan, Ann Arbor, Michigan, United States of America, 5 Taubman Health Sciences Library, University of Michigan, Ann Arbor, Michigan, United States of America, 6 Department of Obstetrics and Gynecology, University of Michigan, Ann Arbor, Michigan, United States of America, 7 Department of Nursing and Midwifery, Lira University, Lira, Uganda

☯ These authors contributed equally to this work.
‡ EA, GKR, RG, FEJ-D, EK, TE, AGA and YRS also contributed equally to this work.
* jbuser@umich.edu

**Data Availability Statement:** The search strategies can be accessed in the repository at https://hdl.handle.net/2027.42/191720.

**Funding:** This work was supported by the Center for International Reproductive Health Training at the University of Michigan (CIRHT-UM) to YS. The

## Abstract

Family planning (FP) is an essential component of public health programs and significantly impacts maternal and child health outcomes. In Uganda, there is a need for a comprehensive review of the existing literature on FP to inform future research and programmatic efforts. This scoping review aims to identify factors shaping the use of FP in Uganda. We conducted a systematic search of eight scholarly databases, for qualitative studies on FP in Uganda. We screened the titles and abstracts of identified articles published between 2002–2023 and assessed their eligibility based on predefined criteria. We extracted data from the 71 eligible studies and synthesized the findings using thematic analysis and the Ecological Systems Theory (EST) individual, interpersonal, community, institutional, and policy-level determinants. Findings reveal the interplay of factors at different socio-ecological levels influencing family planning decisions. At the individual level, the most common determinants related to the EST were knowledge and attitudes of FP. Interpersonal dynamics, including partner communication and social support networks, played pivotal roles. Community-level factors, such as cultural norms and accessibility of services, significantly influenced family planning practices. Institutional and policy-level factors, particularly a healthcare system's quality and policies, also shaped use. Other themes included the intersection of HIV/AIDS on FP practice and Ugandan views of comprehensive abortion care. This scoping review underscores the intricate socio-ecological fabric shaping FP in Uganda. The findings highlight the need for targeted interventions to increase knowledge and awareness of FP, improve access to services, and address social and cultural norms that discourage contraceptive use. Policymakers and program implementers should also consider gender dynamics and power imbalances in FP programs to ensure they are equitable and effective.

funders had no role in study design, data collection and analysis, decision to publish, or preparation of the manuscript.

**Competing interests:** The authors have declared that no competing interests exist.

## Introduction

Family planning (FP) is an essential component of public health programs and has a significant impact on maternal and child health outcomes by reducing high-risk pregnancies and allowing sufficient time between pregnancies [1]. Like many low- and middle-income countries (LMICs), Uganda faces a range of challenges in implementing and scaling up FP programs and comprehensive abortion care (CAC) services. In Uganda, more than half of pregnancies are unintended, and nearly a third of these end in abortion [2]. The estimated unintended pregnancy rate is 145 per 1000 women aged 15–49 years [3]. In recent years, there has been an increasing interest in understanding the determinants and barriers to FP uptake in Uganda, and a few previous studies have been conducted to explore these issues [4–7]. However, there is still a need for a comprehensive review of the existing literature on FP in Uganda to understand the factors that influence FP uptake to inform future research and programmatic efforts.

This scoping review of qualitative studies aims to provide an overview of the current state of knowledge, attitudes, and practices of FP and CAC in Uganda. By focusing on qualitative research, this scoping review seeks to provide a deeper understanding of the social, cultural, and economic factors that shape FP decision-making and CAC in Uganda. The *Ecological Systems Theory* (EST) has been used to identify factors that influence the adoption and dissemination of FP, how new methods or technologies are introduced, and how social and cultural norms influence the uptake of FP practices. The landscape of FP in Uganda is intricate, influenced by a multitude of factors spanning individual, interpersonal, community, institutional, and policy levels.

The EST provides a comprehensive framework to identify the complex interactions shaping FP practices. Over the past decade, Uganda has witnessed evolving dynamics in reproductive health, necessitating a nuanced understanding of the socio-ecological factors at play [7–9]. At the individual level, knowledge, attitudes, and socio-demographic characteristics significantly impact decisions, while interpersonal relationships and social networks within communities play crucial roles. Cultural norms, values, and the accessibility of FP services at the community level further shape adoption patterns. Institutional factors, such as the quality of the healthcare system, and national policies constitute the broader context influencing FP practices. By exploring these interconnections, this review aims to identify patterns, gaps, and inform targeted interventions, contributing to improved reproductive health outcomes in Uganda.

The EST has been used extensively to study reproductive health behaviors [10, 11], including FP uptake [12, 13] and abortion [14]. Studies have shown that attitudes, knowledge, and cultural norms are all significant predictors of FP behavior [15, 16]. For example, individuals with positive attitudes toward FP are more likely to use contraceptives [17, 18]. Meanwhile, individuals who perceive that their social network supports FP are more likely to use contraceptives [19], while those who perceive social disapproval are less likely to do so [20]. Finally, individuals who perceive that they have control over their reproductive health are more likely to use contraceptives than those who perceive barriers to access [21].

This scoping review will use the EST as a guiding framework to explore qualitative studies on FP in Uganda to identify gaps and patterns across the socio-ecological spectrum. By synthesizing the findings of qualitative studies using the EST, we aim to provide a comprehensive understanding of the factors that influence FP uptake in Uganda. This knowledge can be used to inform the development of targeted interventions to increase FP uptake and CAC, improve reproductive health outcomes in Uganda, and provide recommendations for future research and programmatic efforts.

## Methods

### Study design

The study follows the methodological framework Arksey and O'Malley developed for scoping reviews to identify the research question and relevant studies, select studies, chart the data, and report the findings [22]. When one is mapping and exploring the literature on a topic, a scoping review is most appropriate [23, 24].

### Literature search strategy

A comprehensive literature search was conducted using eight scholarly databases: MEDLINE (via Ovid interface), EMBASE (via Embase.com), Scopus, CINAHL (via EBSCOhost), Web of Science Core Collection (via Thomson Reuters), Global Health (via CABI), PsycINFO (via EBSCOhost) and Women's Studies International (via EBSCOhost). Keyword and controlled vocabulary search terms were used to represent concepts related to sexual and reproductive health in the context of FP or CAC in Uganda. The search was conducted by a health sciences informationist (GKR) in May 2022 then updated in May 2023. The search strategies can be accessed in the repository at https://hdl.handle.net/2027.42/191720.

Geographic search terms were used to focus search retrieval on articles referencing Uganda at the country level, by district [25] or capital city of Kampala. Lastly, a revised qualitative/mixed methods search filter was used in all eight database searches [26]. Two unique qualitative/mixed methods search filters were revised for use in Ovid Medline to strive to maximize retrieval of qualitative studies [26, 27]. Final search strategies were determined through test searching and the use of search syntax to enhance search retrieval. No language limits were applied.

A search was conducted in May 2022, followed by an update of search results in May 2023. Search results were limited to articles published from 2002 to 2022, resulting in 4,217 citations exported to EndNote for processing and removal of duplicate citations. A final count of 1,422 citations were assessed and screened in Rayyan [28] according to inclusion and exclusion criteria.

### Inclusion and exclusion criteria

The inclusion criteria for this study were articles that: (1) focused on FP and CAC research studies conducted in Uganda; (2) used qualitative research methods; (3) were published in peer-reviewed journals; and (4) reported the views of male and female Ugandan citizens, healthcare providers, or policymakers. The search included full-text articles published in any language from 2002 to 2023. The timeframe was chosen to identify recent literature and to capture the period after the cessation of the long-running conflict in northern Uganda between the Lord's Resistance Army and the Ugandan Government which lasted for over two decades, causing immense suffering and displacement for the people in the region [29].We included studies with data from multiple countries if Uganda-specific data were reported separately. The exclusion criteria were articles that: (1) focused solely on quantitative research methods or encompassed other publication types (editorials,, protocol papers, or commentaries, because they typically do not present qualitative data; dissertations because they are not peer reviewed; and abstracts because they are not full-length articles); (2) were not relevant to FP or CAC in Uganda; (3) were not published in peer-reviewed journals; and (4) reported the views of non-Ugandan citizens (i.e., refugees from other countries).

### Study selection and data extraction

Two independent reviewers screened the titles and abstracts of the identified articles to determine eligibility for inclusion using the web-based tool Rayyan [30]. A third reviewer resolved

discrepancies. Full-text articles of the selected studies were retrieved, and reviewers from the reviewing team further assessed them for eligibility. The reviewers communicated regularly to achieve consensus about the selection of studies.

Data were extracted from the selected articles by six reviewers using a standardized Microsoft Excel data extraction form that included the author(s), article title, year of publication, study population, study aims, sample size, key findings related to the EST determinants (individual, interpersonal, community, institutional, and policy level), and implications for FP policy and practice in Uganda (S1 Appendix). The quality of the selected articles was assessed using the Critical Appraisal Skills Programme (CASP) tool for qualitative studies [31]. Hand-searching of reference lists was not performed. A deductive approach was used in presenting the data.

## Results

After screening and full-text review, 71 articles were included in the scoping review. Fig 1 shows the PRISMA diagram produced using the PRISMA Flow Diagram tool [32]. Sixty-four of the articles focused on FP. Seven specifically focused on CAC. Fig 1 illustrates the number of articles by publication year of the included FP studies. The earliest FP study was published in 2005, and the most recent one in 2023. A majority of FP studies retrieved from the search results were published between 2013 and 2023 (Fig 2). Fig 3 indicates the characteristics of FP study participants. Adults and women are the most common study population types, with 32 studies including adults and 51 studies including women specifically (Fig 3). Twenty-one articles focused on adolescents. Studies also identified study populations of key stakeholders (n = 9) and healthcare workers (n = 10) (Fig 3). Focus group discussion (FGD) (n = 34) and in-depth interviews (IDIs) (n = 35) were the most common data collection types (Fig 4). Key informant interviews (KIIs) were used in 12 studies, and seven studies used semi-structured interviews (SSIs) (Fig 4). Overall, the quality of the included studies was moderate to high, with most studies meeting the majority of the CASP [31] criteria. The studies in this scoping review covered a broad range of primary aims related to FP in Uganda (Fig 4). Fig 5 presents a comprehensive overview of the EST for FP in Uganda, illustrating the interconnectedness of factors at various socio-ecological levels. At the center of the framework is the individual, influenced by factors such as knowledge, attitudes, and socio-demographic characteristics. Interpersonal relationships and social support networks within communities surround the individual, impacting FP decisions. Cultural norms, values, and the accessibility of services at the community level further shape practices. The outermost layer comprises institutional and policy factors, including the quality of healthcare services and national policies, which provide the broader context for FP practices. Understanding these interactions is crucial for developing targeted interventions to enhance reproductive health outcomes in Uganda.

### Individual level factors

**Knowledge and attitudes.** Several studies highlighted the significance of individual-level factors in FP decisions. Knowledge and attitudes towards contraception emerged as critical determinants of FP use. Individuals with accurate information and positive attitudes were more likely to adopt and sustain FP practices. The main barriers to FP uptake were a lack of knowledge and awareness. In the context of this review of qualitative studies, attitudes refer to an individual's overall positive or negative view toward FP. The most common theme of the findings in the studies (the focus of thirteen studies in total) centered on the individual's reasons to use (or not use) contraceptives. Individuals were more likely to use FP when they desired better child spacing but could be wary of the possible side effects and health concerns

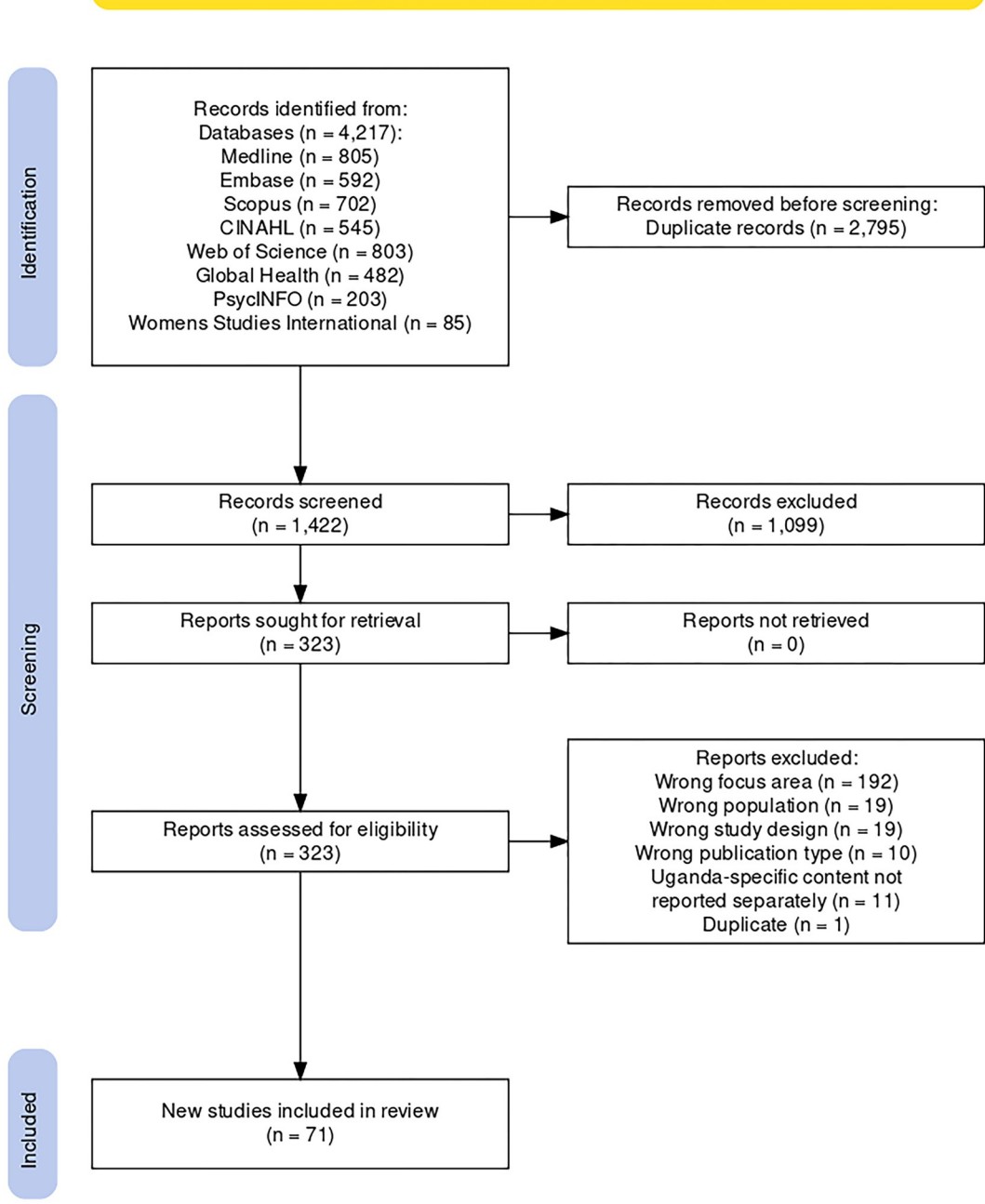

**Fig 1. PRISMA.**

[33–40]. Men often feared that the side effects of contraception methods would hinder sexual activity [38, 41]. People are more interested in using and continue to use methods they are familiar with [36]. The FP method's efficacy is also considered, as is the cost [30]. Individuals were also concerned with their level of privacy [40, 42–44] and would be more likely to use contraception when not having to involve a health care provider [37, 45, 46]. Men who agree to use condoms do so to please their partners, protect themselves and their families from HIV, and in some cases, have multiple partners [47]. Using contraception can foster a sense of

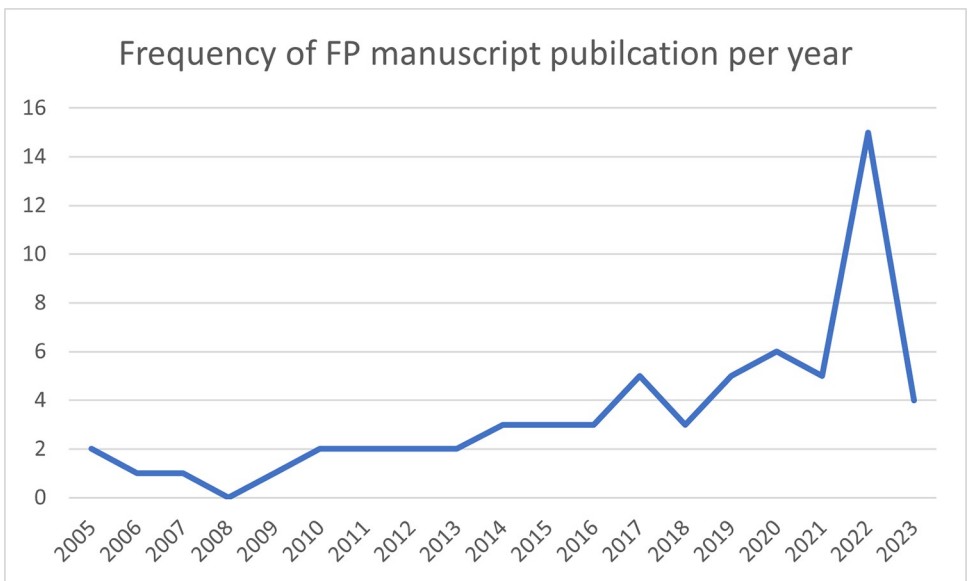

**Fig 2. The publication years included in this scoping review of Ugandan family planning uptake.**

accomplishment in women [48]. Table 1 displays representative quotes from articles organized by EST factors.

Several articles focused on adolescents' knowledge and use of contraceptives. Kyegombe et al. [49] found that, while the specific age varies, the concept of childhood must be protected, including from engaging in sexual activity. Parents who did not support adolescent contraception worried it promotes promiscuity and infertility [50]. Mbalinda et al. [51] and Nobelius et al. [52] found that cultural norms could dissuade teens from using condoms; the older adolescents used them specifically to prevent pregnancy. Other articles [43, 53, 54] also found that

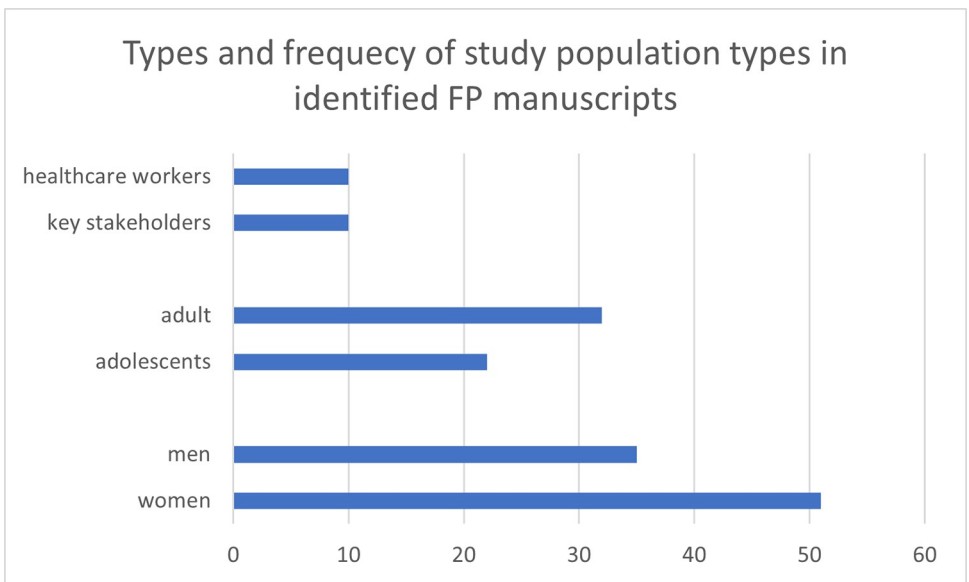

**Fig 3. Characteristics of study participants.** Each study may include participants of both genders (male and female), ages (adolescents and adults), and also have an additional participant role (key stakeholders and healthcare workers).

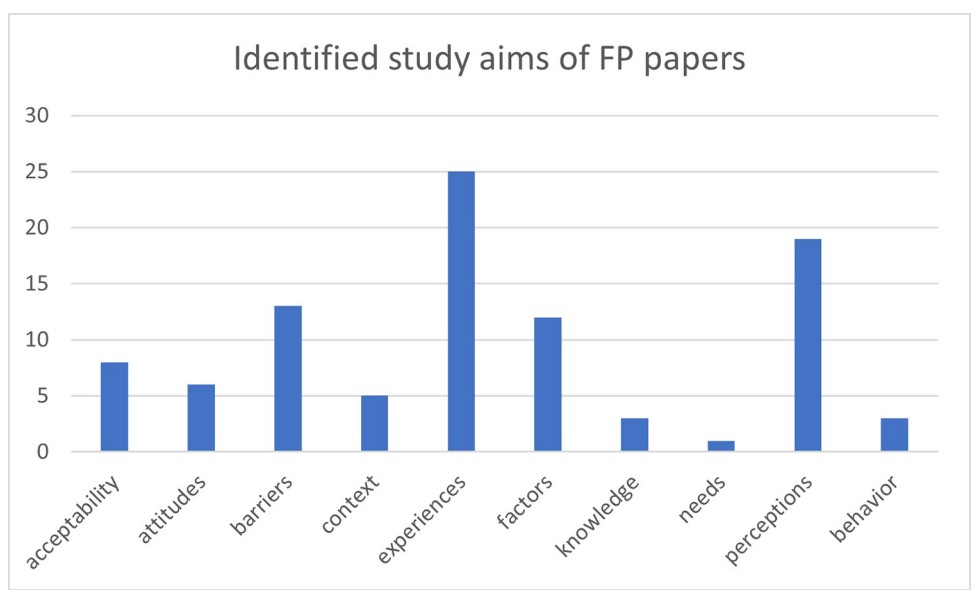

**Fig 4. Study primary aims.** The aims are drawn from the most common words used by authors in their title or aims description, such as knowledge, experiences, and expectations related to family planning. Studies with multiple primary aims are counted multiple times in the figure.

the use of contraceptives among adolescents depended on the information and sources available and how acceptable contraceptives were in the community. Some parents felt unequipped to discuss FP options with their children and instead avoided the topic [50].

## Interpersonal level factors

**Social support networks.**   The influence of social support networks was evident, with family, friends, and colleagues playing roles in shaping FP choices and how much they approved or disapproved of it [40]. For example, people reported religious leaders and mothers-in-law mainly discounting the use of FP [60] and adolescents were discouraged from using self-injective contraception over safety concerns [37]. Parents who did support their adolescent children cited its use to prevent STDs, unwanted pregnancy, and early school dropout [50]. Attitudes of their peers [46, 60] and male partners [46, 61, 62] also influenced FP uptake.

**Male involvement and communication.**   Barriers to FP uptake stemmed from cultural norms prioritizing imparting FP knowledge to adult women over men and adolescents. Several studies [6, 51, 58, 63, 64] reported that men indirectly learned about FP methods from their partners' experiences and the knowledge their partners received from healthcare providers. Cultural and gender norms in Uganda negatively impacted men's opinion of FP, which was considered as the domain of women [63, 65]. Most fears of FP came from misinformation related to the possibility of infertility [66]. When facts werepresented, people gained confidence in discussing options with their partners and healthcare providers [67–69]. A telehealth package studied by Kamulegeya et al. [70] enabled men to share FP info based on informative and timely messaging, which built self-confidence in their knowledge on FP.

## Community level factors

**Cultural norms and values.**   A person's use of contraception can be influenced by the community's gender and social norms around expectations of fertility choices [71]. Traditional

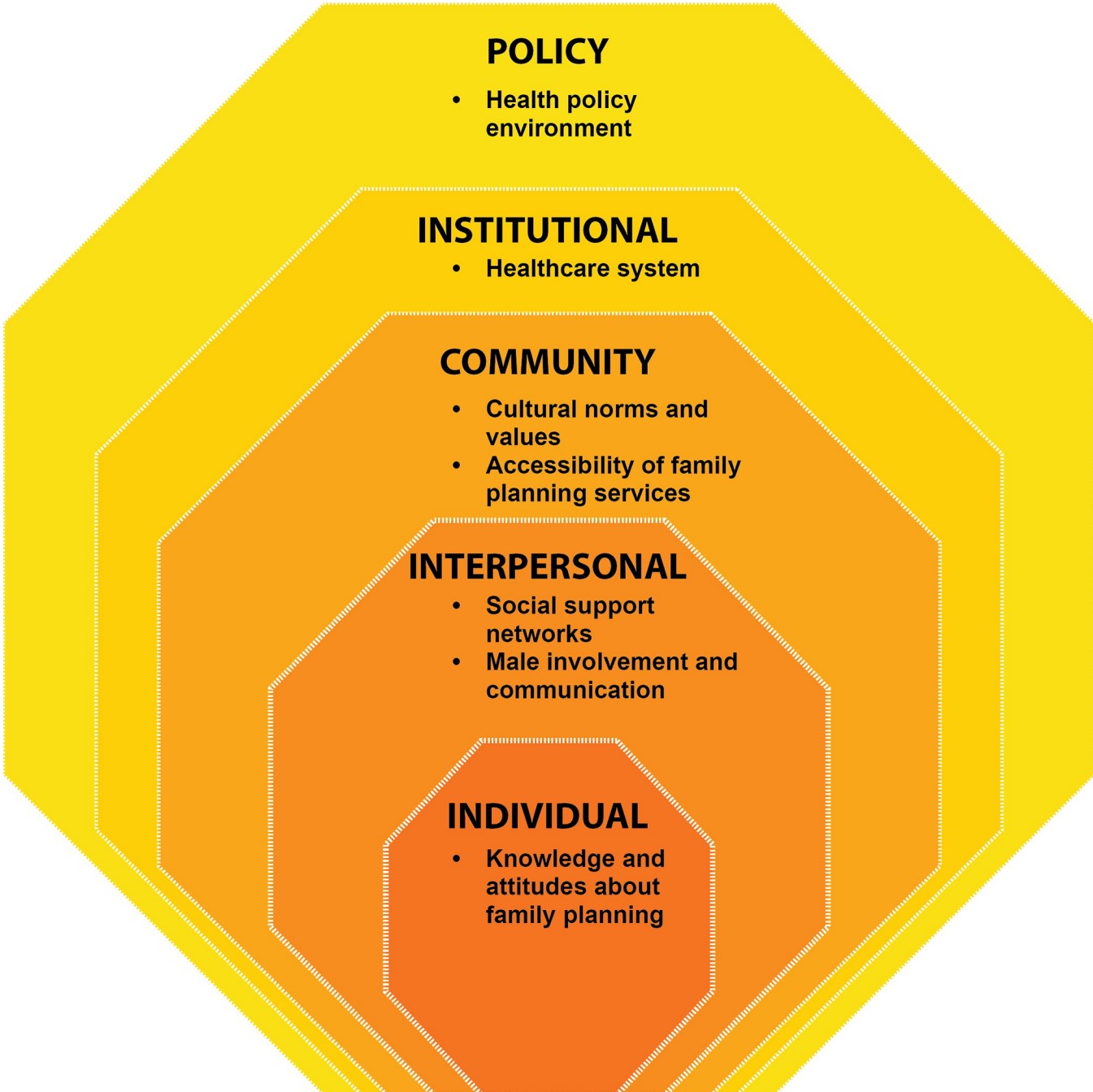

**Fig 5. The ecological systems *Theory* applied to family planning in Uganda.**

Ugandan culture values having several children [60]. Kabagenyi et al. [38] mentions that continued births are a sign by wives of their respect and love for their husbands. These pressures can create a situation where women lack the agency to make decisions about contraception [40]. Willcox et al. [72] found that social-cultural pressures have even more influence in low-income settings. Gaffikin and Aibe [73] found an integrated approach to FP that affirms its importance to the community by demonstrating the links to improved economics and natural resource management.

**Table 1. Representative quotes from articles organized by Ecological Systems Theory levels and factors.**
FGD = focus group discussion, FP = family planning.

| Individual | |
|---|---|
| **EST factors** | **Representative Quote** |
| **Knowledge** | "Women listen to a lot to myths. I heard some saying that when one is on family planning their sexual urge gets interrupted and their husbands may leave them due to that insensitivity to sex." (Female, age 42, IDI) [6] |
| | "Those pills are dangerous; they go through the fallopian tube and go to that area where eggs come from. So, when the pill falls in the middle of all the eggs, it burns them all. . . . .they burn the entire woman's eggs and form a big scar. You may die without ever becoming pregnant". (married woman, 15–19 years, FGD) [53] |
| *Attitudes* | "I cannot practice family planning because I want as many children as I can give birth to. My mother in-law says she wants me to bear many children. Also, God said we should give birth and fill the world [so] why should I limit myself? I want to give birth to 12 children. So far, I have only three children. I give birth to my children here at home and not in the hospital". (Female age 19, IDI, Acholin village) [34] |
| | "Women listen to a lot to myths. I heard some saying that when one is on family planning their sexual urge gets interrupted and their husbands may leave them due to that insensitivity to sex." (Female, age 42, IDI) [6] |
| ***Interpersonal Level*** | |
| Social support networks | "A key challenge to youth access is that. . . there is a fear of reversibility. 'If I use an IUD, how will I get pregnant again?' 'If I use an implant, how will I get pregnant again?' So people don't want to use those long methods because they are scared that if they do, they will never get pregnant. This is a big source of concern for young people. This is a big concern here because once you get married, people expect you to get pregnant." (International NGO representative, IDI) [55] |
| *Male involvement and communication* | "Men in this village do not like to use family planning and they prohibit their wives from using it. So women who come for family planning—they hide from their partners. Men know how and where to check especially for those who use implants they know the right arm to check.... Men know that the implant is put on the right arm and they know the position so they check their wives to find out whether they went secretly to use family planning." (31-year-old female nurse, IDI) [6] |
| Community Level | |
| Cultural norms and values | "A woman should be obedient by listening to her husband. She should also be respectful. She should care for the children and [be] hardworking in her home. She must be humble in her talking, faithful to her husband, and welcoming". (Female, age 19, IDI) [56] |
| *Accessibility of FP services* | *"At the Health Centre the long-term methods are not available. We usually wait for an announcement by Blue Star (a local program that provides long-acting contraceptive methods) that the services are being brought, and that's when we come to the health center."* (Young female client at rural health center, FGD) [45] |
| Institutional and Policy | |
| Healthcare system | "I always talk to the health provider and she finds a way of helping me. I cannot take a decision on my own regarding these challenges. When I got the injection and experienced problems, I came back and talked to her. She told me 'such things happen at the start but you will be fine after some time', and I indeed I got well after." (Female, age 26, IDI) [57] |
| | I did not tell anyone about my side effects with implants [stopped using it] except the health providers because people can spread rumours and yet my husband does not want me to use it." (Female, age 25, IDI) [57] |
| Health policy environment | "For us men, we really like to bring those services nearer to us because the women come here [to the health centre] for antenatal clinics and when they bring children to be immunized they are taught about family planning. Yet we, who don't bring children to be immunized, don't get that information of the methods." (male, FGD) [58] |
| | "New [contraceptive] users were even more affected. They thought that family planning services were also locked-down." (Female health worker, KII) [59] |

**Accessibility of family planning services.** The availability and service delivery of contraception also influences people's ability to use them [45, 71, 74]. The availability of multiple FP options from which to choose increases the uptake [75]. Stockouts are often reported by patients and healthcare providers [40], while policymakers were not aware of the extent of the issue [76]. Women who choose self-injections to maintain privacy, and are aware of the safety risks, will opt to properly dispose of the injections with a healthcare worker, even if more convenient options are available [77]. There remain limited options for male-directed FP methods (including vasectomies) [63], but men believe if more options were available, men's positive attitude towards FP would increase [78].

Fortin et al. [79] found that structural inequalities and health vulnerabilities interact at the intersection of women's identities (derived from their motherhood, marriage status, employment status, education level, and if they had a disability or a chronic illness). More rural areas experience challenges in accessing FP services, which also deters contraception use [34]. The disruption of COVID-19 made it even harder for people to access FP services. People experienced unsteady employment/income, unintended pregnancies, unreliable transportation, and service delivery interruptions [59, 80]. Lockdowns exacerbated the impact of poverty and gender inequality on FP access [59].

## Institutional and policy level factors

**Healthcare system.** Providers play a role in the perception and delivery of sexual and reproductive health services. Providers want to incorporate better FP into their practice and give knowledge to their patients [51, 81, 82]. However, the availability of providers limits their ability to provide health services [51, 83]. More services could be provided with more training [81, 84, 85], and healthcare providers must be supportive to endear trust from their patients [86]. More community outreach in public health interventions may counteract the misinformation [87].

Other articles focused on the role of healthcare workers in educating the community on FP. Kibira et al. [57] and Mbalinda et al. [51] found that healthcare providers were the most influential educational sources about FP. Namanda et al. [60] found that their role was even more influential for married individuals. Healthcare providers' sensitivity to their patients' misconceptions and expectations shape the continued use of contraception [44, 83]. If they exaggerate perceptions of side effects, especially in comparison to the risk of pregnancy, patients are less likely to use contraception [84].

**Health policy environment.** In terms of policies, some of the articles focused on how policies impacted access to FP services. Kaida et. al. [58] discussed the perception and attitudes of Ugandan men towards family planning, considering the Ugandan National Population policy, the objective of which was to increase men's participation in family planning. The study found that FP services were still not targeted towards men, limiting their access to knowledge, but men are willing to be involved in discussions about FP [58]. Tuhebwe et. al. [87] also found that while the intentions of the Uganda reproductive health services standards were to increase access to adolescents, the implementation was underutilized as designed, looked at through the framework of the WHO global standards [88]. Providers designed interventions targeting a wide age group of adolescents and were facility-based, while the adolescents in the study preferred community-based services targeted towards a narrower age range. Grindlay [76], focusing on the frequency of stockouts of FP materials, found that policy makers were unaware of the magnitude of the issues faced by providers. Two studies also focused on the negative effects of the COVID-19 mitigation measures instituted by the Ugandan government during the pandemic on access to FP services. The lockdown decreased family income to pay for services, and limited transportation to travel to village health teams for services [59, 80]. They also magnified pre-existing inequalities in access [59].

In addition to the EST factors, other themes emerged from the included studies: the intersection of HIV/AIDS on FP practice and Ugandan views of CAC.

## Intersection of HIV/AIDS on family planning practice

Nine articles discussed the intersection of HIV/AIDS in FP practice. Communication between partners is still important [62, 89, 90]. Kosugi et al. [89] observed that those who perceived they were at a higher risk of HIV/AIDS were more likely to use dual-method contraceptives. HIV prevention can be confusing, and people struggle to balance their perceptions of HIV risk with their desire to have children [91–93]. Health providers struggle to provide integrated HIV, antenatal, and prenatal care services [94]. Testing for HIV is difficult due to a fear of the results and the perceived social stigma [95]; though self-testing may keep the information confidential, they are not as accurate, and a positive result could still prove harmful [96].

## Comprehensive abortion care

Seven of the articles focused on the perceptions of comprehensive abortion care. Of the seven articles, two articles focused on men as a study population, four on women, and one on healthcare workers. The primary data collection type was IDIs, with all 7 studies performing IDIs as one of their data collection methods. Only one article [97] used SSIs and FGDs. The publication years also varied. Two articles were published in 2022, four were published between 2005 and 2017, and one article was published in 2023. The primary aims of the articles were also varied, focusing on perceptions, experiences, and attitudes toward comprehensive abortion care.

The perception of abortion care is influenced by the individual's attitude towards it. Women decide abortion is necessary due to financial constraints, unplanned pregnancy, and complicated social networks [98]. They may desire to keep their decision to have an abortion private, but that can lead to unsafe abortions and late care-seeking [99]. Moore et al. [100] found that men perceived women's reasons for seeking abortions differently from women. B. Nyanzi et al. [101] also found that men's views on abortion are ambivalent, seeing abortions as either a solution or something to be avoided.

Gender and social norms also influence people's views of abortion care. The agency to make such a decision is constrained by gender norms [99]. Moore et al. [100] found that men expressed more rigid anti-abortion sentiments than they actually felt. Women often consult many people in their community before deciding on how to access abortion services [102]. Kabunga et al. [98] found that women who have an abortion experience a loss of family support and internalized perceived stigma.

Perceived behavior control guided by the health care structures also influences views on abortion care. In terms of the availability of healthcare workers, midwives were viewed as competent and more present at facilities, however doctors were still viewed as needed by women in case of emergencies [103]. The perception of abortion care treatment is informed by the outcomes experienced by patients. Treatment with misoprostol to manage incomplete abortions was viewed positively when it was successful, and the women felt safe [104]. Such treatment also received unanimous support from healthcare workers because it was deemed safe, effective, and inexpensive, but it does put a strain on the healthcare facility and staff [102]. Their satisfaction decreased when they experienced side effects, such as worrying bleeding [104].

## Limitations of review articles

In this review, we identified several limitations in the studies analyzed. Firstly, social desirability bias may have influenced participant responses due to concerns about providing socially acceptable answers [46, 75, 77, 99, 103] or lack of privacy during data collection [40], especially

when sensitive questions were asked [49]. Secondly, recall bias might have been present, as participants may have had difficulty accurately recalling events that occurred years ago, potentially leading to inaccuracies in the data [51, 67]. Additionally, a limitation in the studies was the use of small sample sizes, which could affect the generalizability of the findings to larger populations or different contexts [41, 61, 69, 83, 85]. Moreover, some studies were geographically limited [96], conducted in small areas in Uganda, restricting the applicability of results to broader populations or other regions. Finally, a few studies relied on a limited number of interviews [62, 89], possibly compromising the comprehensiveness and depth of insights obtained for the research topic. These weaknesses highlight areas for improvement in future research and call for careful interpretation of the findings.

## Discussion

The purpose of this scoping review of qualitative studies conducted in Uganda was to identify the socio-ecological factors shaping the use of FP and CAC in Uganda. The scoping review aimed to map the existing literature on FP in Uganda, identify key themes, and explore the gaps and future directions for research. The findings illuminate the complex interplay of individual, interpersonal, community, institutional, and policy-level factors that shape FP decision-making in Uganda. By understanding the diverse range of influences across different socio-ecological levels, stakeholders can develop tailored interventions that address specific barriers and promote informed decision-making.

At the individual level, our review highlights the pivotal role of knowledge, attitudes, and socio-demographic characteristics in shaping FP utilization. Educating individuals about FP methods and debunking misconceptions is essential for promoting informed decision-making. Moreover, understanding how socio-demographic factors such as age, education, and income intersect with FP practices can inform targeted interventions aimed at reaching vulnerable populations. Attitudes towards FP were generally positive among both men and women. However, negative attitudes towards FP were also reported, primarily by men, highlighting the need for targeted interventions to address misconceptions and provide accurate information about FP methods. Interpersonal dynamics emerged as critical determinants of FP behavior. Effective communication within couples and the presence of supportive social networks significantly influence contraceptive uptake. The same holds for research performed in other LMICs where factors associated with the unmet need for FP and non-contraception use are common across different settings [105]. In a synthesis of systematic reviews of factors influencing contraception choice and use globally, D'Souza and colleagues [106] found that factors affecting contraception use are similar among women globally. Use of FP is influenced by relationship status, women's knowledge, beliefs, and perceptions of side effects and health risks, along with male partners, peers' views, and families' expectations, all having a strong influence [106]. Strengthening these relationships through counseling and community-based initiatives can enhance FP decision-making processes.

Community-level factors, including cultural norms and accessibility of services, profoundly impact FP practices. Cultural norms, or the influence of social networks on FP decision-making, were found to be an important factor in contraceptive use. Adolescents' knowledge and use of contraceptives also factored predominantly in research performed in Uganda. Adolescents in Uganda are not alone in their unmet need for FP. Chandra-Mouli et al. [107] assert that all adolescents in LMICs have obstacles accessing their right to contraception, and countries should remove social and medical restrictions to delivering preferred contraception to adolescents. Addressing cultural barriers and ensuring the availability of FP services in remote and marginalized communities are imperative for promoting reproductive health equity.

Moreover, community engagement strategies that involve local leaders and stakeholders can foster a supportive environment for FP uptake.

Social support from spouses, family members, and community health workers increased the likelihood of FP uptake in Uganda. However, social norms around gender roles and male dominance in decision-making were identified as barriers to contraceptive use. In a non-Uganda-specific systematic review, Mandal and colleagues [108] evaluated gender-integrated FP and maternal health interventions in LMICs and proposed that gender constructs, such as gender-equitable attitudes and decision-making power, must be adapted to examine how empowerment and improvements in gender-related factors can produce positive FP outcomes.

Institutional and policy-level factors play a crucial role in shaping the FP landscape. Barriers to accessing FP methods, such as lack of knowledge, limited access to services, and cost, were reported by participants. Addressing these barriers through improved access to FP information and services would be important in increasing perceived control over reproductive health decisions in Uganda and beyond. L. M. Williamson et al. [109] conducted a review of qualitative research to examine the limits to modern contraceptive use identified by young women in developing countries and determined that increasing modern contraceptive method use requires community-wide, multifaceted interventions, and the combined provision of information, life skills, support and access to youth-friendly services. Improving the quality and accessibility of healthcare services, as well as advocating for supportive policies, are essential for enhancing FP access and utilization. Additionally, efforts to strengthen healthcare infrastructure and increase funding for FP programs are vital for sustaining reproductive health initiatives in Uganda.

In addition to the EST constructs, other important themes emerged from the included studies, such as the intersection of HIV/AIDS on FP practice and Ugandan views of CAC. These findings highlight the need for targeted interventions that involve healthcare providers and address gender norms and dynamics within communities. Similar to studies in Uganda, a review of evidence about meeting the RH needs of key female populations affected by HIV in LMICs found that restrictive policy environments, stigma and discrimination in health care settings, gender inequality, and economic marginalization restrict access to services and undermine the ability to achieve reproductive intentions safely [110]. Meanwhile, a systematic review of the contraceptive and abortion knowledge, attitudes, and practices of adolescents in LMICs to increase the understanding of the sexual and reproductive health dynamics that they face suggests severe limitations in the access to safe and effective methods of contraception and safe abortion services [111].

The findings from this scoping review also suggest several gaps in the existing literature on FP in Uganda. While the role of men in FP decision-making was explored in several articles, there was a limited focus on vasectomy as a FP method. Most studies were conducted in urban areas. More research is needed to understand the views of those living in rural parts of the country. The influence of cultural and religious beliefs on FP behaviors also warrants further investigation. Additionally, there were limited studies about using newer technologies, such as mobile phones, social media, and telehealth, to improve reproductive health and FP.

## Application to programing, service provision, and policy

By understanding the diverse range of influences across different socio-ecological levels, stakeholders can develop tailored interventions that address specific barriers and promote informed decision-making. One key area where the findings can impact programming is in the development of tailored interventions. Understanding the unique socio-demographic characteristics and knowledge gaps among different demographic groups allows for the design of targeted interventions that address specific needs. Furthermore, recognizing the importance of

interpersonal communication in FP decision-making suggests the need to strengthen communication skills within couples and promote supportive social networks. This could involve implementing couples counseling sessions, establishing community-based support groups, or initiating peer-to-peer education programs.

Moreover, addressing cultural norms and values is essential in promoting FP and CAC services. By engaging with local communities to challenge harmful stereotypes and myths surrounding contraception and abortion, programs can foster a supportive environment for reproductive health decision-making. Additionally, ensuring the accessibility and quality of FP and CAC services is crucial for promoting uptake. Programs can focus on expanding service availability in underserved areas and reducing barriers such as cost, distance, and stigma.

Leveraging evidence from this review can support advocacy efforts for policies that promote reproductive health and rights. Policy programming can also be used for strengthening healthcare infrastructure and implementing policies that safeguard individuals' access to comprehensive reproductive healthcare services.

## Strengths and limitations

A strength of this study is that it is the first comprehensive scoping review of qualitative literature in Uganda focused on FP that the authors are aware of. The many databases searched allowed us to capture a wide range of studies focused on FP. In terms of limitations, hand searching or reviewing grey literature sources may have increased the number of articles discovered. Furthermore, the studies included in this review were conducted only within Uganda, which may limit the generalizability of the findings outside the country.

## Conclusions

Overall, this scoping review provides a comprehensive overview of the existing literature on FP in Uganda, using the EST as a guiding framework, and identifies key themes and gaps for future research. By addressing factors at multiple levels, including individual, interpersonal, community, institutional, and policy levels, stakeholders can develop holistic interventions that promote reproductive health. The findings highlight the importance of addressing attitudes, cultural norms, and behavior in increasing FP uptake and improving reproductive health outcomes in Uganda. There is a clear need for comprehensive interventions that address socio-cultural norms, expand access to information and services, and tackle structural barriers to SRHR across various contexts.

The findings from this scoping review can be used to inform future FP programming and policy in Uganda, with the potential to improve the accessibility, quality, and utilization of FP services, particularly among marginalized populations. Future research should address the identified gaps in the literature, such as vasectomy as an FP option and the influence of cultural and religious beliefs on FP behaviors, especially in rural areas. An exploration of Ugandan views on newer technologies to improve reproductive health and FP is also warranted. Targeted interventions that involve health care providers and address gender norms and dynamics within communities may be vital to increasing FP uptake in Uganda. Moving forward, interdisciplinary collaboration and longitudinal research are needed to advance our understanding of FP dynamics and improve reproductive health outcomes in Uganda.

## Supporting information

**S1 Appendix. Full list of articles included in the scoping review.**
(XLSX)

## Author Contributions

**Conceptualization:** Julie M. Buser.

**Data curation:** Gurpreet K. Rana.

**Formal analysis:** Julie M. Buser, Pebalo F. Pebolo, Ella August, Rachel Gray, Faelan E. Jacobson-Davies, Edward Kumakech, Anna Grace Auma, Yolanda R. Smith.

**Funding acquisition:** Tamrat Endale.

**Methodology:** Julie M. Buser.

**Supervision:** Yolanda R. Smith.

**Validation:** Julie M. Buser.

**Visualization:** Gurpreet K. Rana, Faelan E. Jacobson-Davies, Tamrat Endale.

**Writing – original draft:** Julie M. Buser, Pebalo F. Pebolo, Ella August, Gurpreet K. Rana, Rachel Gray, Faelan E. Jacobson-Davies, Edward Kumakech, Anna Grace Auma, Yolanda R. Smith.

**Writing – review & editing:** Julie M. Buser, Pebalo F. Pebolo, Ella August, Gurpreet K. Rana, Rachel Gray, Faelan E. Jacobson-Davies, Edward Kumakech, Tamrat Endale, Anna Grace Auma, Yolanda R. Smith.

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
