## [Decision Letter · Decision Letter 0]

2 Feb 2024

PGPH-D-23-02496

Scoping review of qualitative studies on family planning in Uganda

Dear Dr. Buser,

Thank you for submitting your manuscript to PLOS Global Public Health. After careful consideration, we feel that it has merit but does not fully meet PLOS Global Public Health’s publication criteria as it currently stands. Therefore, we invite you to submit a revised version of the manuscript that addresses the points raised during the review process.

The manuscript has been evaluated by two reviewers, and their comments are available below. Reviewer #1 has raised concerns about the theoretical framework underpinning your analysis and the interpretation of your findings, whilst Reviewer #2 requests revisions to the Introduction and risk of bias assessments.

Risk of bias assessments are not typically included as part of a scoping review. However, as part of your revisions could you please clarify why you consider that your research question is more suited to a scoping review rather than a systematic review. Although Reviewer #1 has requested more discussion about how results observed in these articles might affect FP and CAC programming, please ensure that your scoping review does not speculate on implications for practice or clinically relevant questions.

We look forward to receiving your revised manuscript.

Kind regards,

Marianne Clemence

Staff Editor

Journal Requirements:

1. We noticed you have some minor occurrence of overlapping text with the following previous publication(s), which needs to be addressed:

- https://doi.org/10.1155/2023/7583550

-https://doi.org/10.1186/1742-4755-11-1

In your revision ensure you cite all your sources (including your own works), and quote or rephrase any duplicated text outside the methods section. Further consideration is dependent on these concerns being addressed.

Additional Editor Comments (if provided):

Reviewers' comments:

Reviewer's Responses to Questions

**Comments to the Author**

1. Does this manuscript meet PLOS Global Public Health’s publication criteria? Is the manuscript technically sound, and do the data support the conclusions? The manuscript must describe methodologically and ethically rigorous research with conclusions that are appropriately drawn based on the data presented.

Reviewer #1: Yes

Reviewer #2: Yes

2. Has the statistical analysis been performed appropriately and rigorously?

Reviewer #1: N/A

Reviewer #2: Yes

3. Have the authors made all data underlying the findings in their manuscript fully available (please refer to the Data Availability Statement at the start of the manuscript PDF file)?

Reviewer #1: Yes

Reviewer #2: Yes

4. Is the manuscript presented in an intelligible fashion and written in standard English?

Reviewer #1: Yes

Reviewer #2: Yes

5. Review Comments to the Author

Reviewer #1: The topic of drivers of family planning and CAC choices is an important one, and this article is systematically researched and analyzed. However, two items impede this valuable research from achieving its maximum impact. First, the theory of planned behavior as an analytical framework underemphasizes the enormous effect of provider influence, structural gender norms, service quality, policy and policy implementation, and systemic barriers to women's mobility and financial means. While a study of individual level influences on FP- and CAC-related choices may lend value it should be analyzed using a framework that gives greater emphasis to the many factors at the health system, sturcural, and political levels. I recommend using the same results but reanalyzing using a more holistic theoretical framework. The second issue is the relevance of the findings. While the discussion was thorough, the application of the results to programing, service provision, or policy was not robust. The article needs a thoughtful discussion of how these results observed in these articles might affect FP and CAC programming.

Reviewer #2: This review focuses on an interesting and important topic around FP in Uganda. The search strategy is sound and the findings have been synthesised well using the TPB constructs to guide themes. A few suggestions to improve the paper are below:

-The introduction would benefit from providing further data and evidence highlighting the prevalence of unintended pregnancy and high risk pregnancy recorded in Uganda. This will show the problem clearer and justify why looking to improve FP in Uganda is required.

- The Methods should include a risk of bias assessment of each article included to ensure added rigour in justifying which studies were included. Using JBI critical appraisal tools for example https://jbi.global/critical-appraisal-tools can be used to assess bias across various types of articles. This is just one resource that can be used. The same two independent reviewers can undertake these assessments and include the risk of bias table as a supplementary file.

6. PLOS authors have the option to publish the peer review history of their article (what does this mean?). If published, this will include your full peer review and any attached files.

**Do you want your identity to be public for this peer review?** For information about this choice, including consent withdrawal, please see our Privacy Policy.

Reviewer #1: No

Reviewer #2: No

---

## [Decision Letter · Decision Letter 1]

15 May 2024

Scoping review of qualitative studies on family planning in Uganda

PGPH-D-23-02496R1

Dear Dr. Julie M. Buser,

We are pleased to inform you that your manuscript 'Scoping review of qualitative studies on family planning in Uganda' has been provisionally accepted for publication in PLOS Global Public Health.

Best regards,

Million Phiri, PhD

Academic Editor

Reviewer Comments (if any, and for reference):

Reviewer's Responses to Questions

**Comments to the Author**

1. If the authors have adequately addressed your comments raised in a previous round of review and you feel that this manuscript is now acceptable for publication, you may indicate that here to bypass the “Comments to the Author” section, enter your conflict of interest statement in the “Confidential to Editor” section, and submit your "Accept" recommendation.

Reviewer #1: All comments have been addressed

Reviewer #3: All comments have been addressed

2. Does this manuscript meet PLOS Global Public Health’s publication criteria? Is the manuscript technically sound, and do the data support the conclusions? The manuscript must describe methodologically and ethically rigorous research with conclusions that are appropriately drawn based on the data presented.

Reviewer #1: Yes

Reviewer #3: Yes

3. Has the statistical analysis been performed appropriately and rigorously?

Reviewer #1: N/A

Reviewer #3: Yes

4. Have the authors made all data underlying the findings in their manuscript fully available (please refer to the Data Availability Statement at the start of the manuscript PDF file)?

Reviewer #1: Yes

Reviewer #3: Yes

5. Is the manuscript presented in an intelligible fashion and written in standard English?

Reviewer #1: Yes

Reviewer #3: Yes

6. Review Comments to the Author

Reviewer #1: This is a strong article, with all comments addressed. The use of the Ecological Systems Theory is effective and serves as the groundwork for a meaningful analysis of behaviors and systems affecting FP and related services.

Reviewer #3: The paper seems to have addressed initial concerns raised. I would just suggest that the misalignment and formatting issues be addressed throughout the paper.

Only concern is on the delineation of the factors using the EST. For the most part, it seems rather arbitrary. For instance, I don't see how the first quote under social support should be interpersonal in a manner that is different from individual factors. This part can benefit from enhanced clarity.

7. PLOS authors have the option to publish the peer review history of their article (what does this mean?). If published, this will include your full peer review and any attached files.

**Do you want your identity to be public for this peer review?** For information about this choice, including consent withdrawal, please see our Privacy Policy.

Reviewer #1: No

Reviewer #3: **Yes: **Simona Simona
